# Signaling Pathways in the Pathogenesis of Barrett’s Esophagus and Esophageal Adenocarcinoma

**DOI:** 10.3390/ijms24119304

**Published:** 2023-05-26

**Authors:** Ksenia Maslenkina, Liudmila Mikhaleva, Maxim Naumenko, Rositsa Vandysheva, Michail Gushchin, Dmitri Atiakshin, Igor Buchwalow, Markus Tiemann

**Affiliations:** 1A.P. Avtsyn Research Institute of Human Morphology, Petrovsky National Research Center of Surgery, 119991 Moscow, Russia; ksusha-voi@yandex.ru (K.M.); mikhalevalm@yandex.ru (L.M.); maxim-naumenko@mail.ru (M.N.); v.rositsa@mail.ru (R.V.); guschin.michail@yandex.ru (M.G.); 2Research and Educational Resource Centre for Immunophenotyping, Digital Spatial Profiling and Ultrastructural Analysis Innovative Technologies, Peoples’ Friendship University of Russia Named after Patrice Lumumba, 117198 Moscow, Russia; atyakshin-da@rudn.ru; 3Institute for Hematopathology, Fangdieckstr. 75a, 22547 Hamburg, Germany; mtiemann@hp-hamburg.de

**Keywords:** Barrett’s esophagus, inflammatory signaling pathways, intestinal metaplasia, mutational load, p53, dysplasia, carcinogenesis, esophageal adenocarcinoma

## Abstract

Barrett’s esophagus (BE) is a premalignant lesion that can develop into esophageal adenocarcinoma (EAC). The development of Barrett’s esophagus is caused by biliary reflux, which causes extensive mutagenesis in the stem cells of the epithelium in the distal esophagus and gastro-esophageal junction. Other possible cellular origins of BE include the stem cells of the mucosal esophageal glands and their ducts, the stem cells of the stomach, residual embryonic cells and circulating bone marrow stem cells. The classical concept of healing a caustic lesion has been replaced by the concept of a cytokine storm, which forms an inflammatory microenvironment eliciting a phenotypic shift toward intestinal metaplasia of the distal esophagus. This review describes the roles of the NOTCH, hedgehog, NF-κB and IL6/STAT3 molecular pathways in the pathogenesis of BE and EAC.

## 1. Introduction

Barrett’s esophagus (BE) is a premalignant lesion that involves the development of intestinal metaplasia in the distal esophagus, and is caused predominantly by long-term exposure to bile reflux. Histological evaluations have revealed several gland phenotypes across the metaplasia segment, arranged in a mosaic fashion [1,2,3,4]. These phenotypes include cardiac, oxynto-cardiac and intestinal phenotypes. Complex histological approaches, immunohistochemical approaches and the sequencing of mitochondrial DNA have revealed that the cardiac phenotype appears earliest and gives rise to all other gland phenotypes in the metaplasia segment during clonal evolution. A high mutational load in BE [5,6], along with marked clonal heterogeneity [4], provides a basis for the development of dysplasia and esophageal adenocarcinoma (EAC).

Two major hypotheses exist regarding the source of BE: the classical mechanism of healing a caustic injury [2,7,8,9] and a phenotypic shift in the context of a so-called cytokine storm [10,11,12,13,14]. These hypotheses are not mutually exclusive and can accompany each other, thereby leading to the reprogramming of stem cells (SCs) and subsequent changes in the architecture of the esophageal mucosa. These changes include a shift in the epithelial type and the recruitment of the inflammatory microenvironment in the lamina propria mucosa and submucosa. This review describes the roles of NOTCH, hedgehog (Hh), NF-κB and IL6/STAT3 signaling in the pathogenesis of BE.

## 2. Gastro-Biliary Reflux as An Inducer of Intestinal Metaplasia

Substantial evidence suggests that gastro-biliary reflux has a major role in BE development, and BE is observed in 2–14% of patients with gastro-esophageal reflux disease [15,16,17]. The pathogenetic association of acid and bile reflux with BE has been demonstrated in studies measuring pH in humans [18,19,20], or using laboratory models of BE [10,12,13,21,22] or cell lines [23,24,25,26,27,28,29,30,31,32,33,34]. Studies using animal models have been unable to fully elucidate the pathogenesis of BE because, after surgery, the exposure of the esophagus to reflux is not near physiological levels. In contrast, studies using cell lines can explain the biological mechanisms of bile exposure but cannot provide reliable evidence on the cellular origins of BE.

Acid reflux results in the formation of a pH gradient along the metaplasia segment that enables optimal bile salt solubility, thus allowing entry of bile salts into epithelial cells [35]. The multiple signaling pathways involved in BE are activated in epithelial cells, thus eliciting epithelial–stromal interactions. Inside epithelial cells in the distal esophagus, bile acids cause injury to organelles, including the mitochondrial membranes, thus triggering uncontrolled generation of reactive oxygen species, oxidative stress and DNA damage. Bile acids also induce and augment the release of proinflammatory cytokines, including IL1β, IL6, IL8, TNF-α [10,12,13,36,37,38], PGE2 [24] and COX-2 [25]. These cytokines activate the NF-κB signaling pathway, thus preventing apoptosis, enhancing the proliferation of epithelial cells, and favoring regeneration after injury. Therefore, the development of intestinal metaplasia is an adaptational mechanism in the distal esophagus. Simultaneously, repeated reflux exposure, DNA injury, including TP53, and the accumulation of multiple mutations and genomic instability, drive the formation of dysplasia and EAC [4,5,6,39,40].

### 2.1. Roles of the NOTCH Signaling Pathway in the Development of Intestinal Metaplasia

Exposure of the keratinocyte cell lines EPC1 and EPC2 to bile acids induces changes in the expression of numerous genes, including genes involved in squamous differentiation, oxidative stress, DNA repair and the cell cycle [29]. Activation of transcription factor CDX2 is the key event in the phenotypic shift toward intestinal differentiation. Of the bile acids, cholic and dehydrocholic acids cause the most prominent activation of CDX2 in cellular cultures of keratinocytes. Moreover, keratinocytes transfected with CDX2 expression vectors show the upregulation of MUC2 transcription, an early sign of cellular reprogramming from squamous to intestinal differentiation [23]. This process is caused by the activation of the NF-κB signaling pathway [23,33] and the inhibition of NOTCH [26,27,28]. The exposure of EAC cell cultures (OE19, OE33) and immortalized squamous epithelium (Het-1A) to bile acids decreases the expression of NOTCH receptors, thereby leading to inhibition of the transcription factor Hes-1, activation of ATOH-1 and direct stimulation of CDX2 expression. Moreover, activation of the NOTCH receptor ligand Dll1 increases the expression of ATOH-1, and high expression of CDX2 inhibits Hes-1, thus leading to fixation of the intestinal phenotype (Figure 1). Notch signaling pathway inhibition in immortalized keratinocytes elicits changes in the morphology of the basal layer of the squamous epithelium and the acquisition of columnar features, including increased expression of CDX2, KRT8, KRT18, KRT19, KRT20, MUC2, MUC3B, MUC5B, MUC17, SOX9, villin and Das-1, and decreased expression of the squamous markers CK4, Tap63, KRT5, KRT13 and KRT14 [31,32].

### 2.2. Roles of Hedgehog Signaling Pathways in the Development of Intestinal Metaplasia

Bile acid exposure of the Het-1A cell line under acidic pH leads to Hh signaling pathway activation, which is normally observed in the esophagus during embryogenesis and is absent in the squamous epithelium. Hh signaling in BE occurs in both the epithelium and stromal elements. The transmembrane receptor PTCH is activated after the binding of Hh ligands and subsequently inhibits the protein SMO, thus leading to the activation of transcription factor Gli and increased expression of different genes [41,42], including SOX9 in the epithelium and BMP4 in the stromal elements. Owing to epithelial–stromal interactions, BMP4 activates pSMAD1/5/8 [43] in keratinocytes, thus leading to the expression of SOX9 [42], which is normally expressed in the colon. The coactivation of the transcription factor CDX2 is necessary for the development of the intestinal phenotype KRT20+ and MUC2+, because CDX2 forms a complex with pSMAD, binds the promoter and induces transcription of MUC2 [44]. In addition, Wang D. H. et al. [45] revealed that the Hh-dependent transcription factor FOXA2 induces the development of the intestinal phenotype without CDX2 activation, through the direct activation of MUC2 expression and an increase in the expression of AGR2, which controls the processing of MUC2 (Figure 2).

Therefore, the development of intestinal metaplasia in the distal esophagus is a complex multiple-stage process determined by the activity of several complementary signaling pathways, including epithelial–stromal interactions. This process does not involve mature epithelial cells, but instead involves SCs and progenitor cells, which give rise to several cell populations. Clonal evolution of these cell lineages leads to the development of BE.

## 3. Possible Cellular Origin of BE

Several possible origins of BE with different pathogenetic mechanisms have been proposed. Chronic exposure of the distal esophagus to bile and acid causes intestinal metaplasia due to cellular reprogramming. This reprogramming involves changes in transcription factor expression and a shift in cellular phenotype. The proposed pathways of cellular reprogramming include transdifferentiation of the squamous epithelium and transcommitment of progenitor cells [40,46,47,48].

Transdifferentiation is a process in which one well-differentiated cellular type (e.g., the squamous epithelium) differentiates into another cellular type (columnar epithelium) [46,49]. In direct transdifferentiation, the change in cellular phenotype does not involve cell division. Indirect transdifferentiation involves initial dedifferentiation of the squamous epithelium to progenitor cells and subsequent acquisition of the new phenotype of metaplastic columnar epithelium. Therefore, indirect transdifferentiation also involves the transcommitment of progenitor cells. The term transcommitment is more precise, because transdifferentiation describes the generation of one cell type from another, whereas in BE, several cellular phenotypes (lines of differentiation) arise from multipotent SCs [50].

Six potential cellular origins of columnar metaplasia in the distal esophagus have been proposed:SCs and progenitor cells of the squamous epithelium;SCs and progenitor cells of the gastro-esophageal junction;SCs and progenitor cells of the submucosal glands and their ducts;SCs and progenitor cells of the first oxyntic gland of the stomach;Residual embryonic cells;Circulating bone-marrow-derived multipotent SCs.

### 3.1. SCs and Progenitor Cells of the Squamous Epithelium

The multilayered epithelium at the squamocolumnar junction, which consists of several layers of immature squamous epithelia covered with mucus-secreting columnar cells, favors the transcommitment of squamous progenitor cells. The multilayered epithelium demonstrates the ultrastructural and immunohistochemical features of both the squamous and columnar epithelia. Scanning electron microscopy revealed that the multilayered epithelium displays both intercellular ridges (a feature of the squamous epithelium); short, stubby microvilli; and bulging mucus (typical of the metaplastic columnar epithelium) [51]. Moreover, the basal cells of the multilayered epithelium co-express CK19 (a feature of the columnar epithelium) and CK4 (typical of the squamous epithelium) [52]. The multilayered epithelium is associated with gastro-esophageal reflux disease and is not present in normal gastro-esophageal junctions [53]. On the basis of these data and their own findings, Chandrasoma P. T. et al. have described a revolutionary concept of cardia defined as a reflux-damaged dilated distal esophagus which is lined with a metaplastic columnar epithelium [54,55,56]. Therefore, the first oxyntic gland histologically demarcates the stomach and the esophagus. The origin of BE from squamous esophageal progenitor cells was also confirmed by Nicholson A. M. et al., 2012 [1], who observed the same mutations in mitochondrial DNA in the metaplastic columnar epithelium and adjoined squamous epithelium.

### 3.2. SCs and Progenitor Cells of the Gastro-Esophageal Junction

The SCs of the gastro-esophageal junction were suggested as a possible origin by Jiang M. et al., 2017 [57], who found basal progenitor cells with a phenotype of p63+ KRT5+ KRT7+ in the multilayered epithelium of the transitional zone in human biopsies and in mice. In mice with ectopic expression of CDX2, these transitional basal progenitors were found to differentiate into an intestinal-like epithelium including MUC2+ TFF3+ goblet cells. These progenitor cells are likely to be the same squamous progenitor cells as those discussed earlier.

### 3.3. SCs and Progenitor Cells of the Submucosal Glands and Their Ducts

The SCs and progenitor cells of the submucosal glands of the esophagus are another possible source of the columnar metaplasia. A thorough histological examination indicated that the multilayered epithelium is a continuation of the submucosal esophageal gland ducts [58,59]. Glickman J.N. et al. [58] revealed the expression of the markers CK7, 8/18, 19 and 20 in the columnar epithelium, squamous epithelium and submucosal glands and their ducts. Moreover, they found a high proliferative index of the multilayered epithelium of ducts in 88% of cases (14 of 16 cases). Therefore, a multilayered epithelium at the surface, which is believed to be the earliest sign of reflux injury, may arise from the SCs of the esophageal submucosal gland ducts.

In pigs, the multipotent SCs of the submucosal glands are involved in regeneration after injury to the squamous epithelium. Moreover, these cells express the columnar epithelial markers SOX9, CK7 and CK8 [60]. Owen R.P. et al. [61] found similar expression of different markers (including LEFTY1 and OLFM4) in the SCs in the submucosal glands and the metaplastic columnar epithelium. These findings implicate the SCs of the submucosal glands as the source of BE. Moreover, the strongest evidence was provided by DNA sequencing that revealed similar mutations in CDKN2A and TP53 (including loss of heterozygosity) in the metaplastic epithelium and the ducts of the submucosal glands [62].

### 3.4. SCs and Progenitor Cells of the First Oxyntic Gland of the Stomach

Evidence has also suggested that the metaplastic epithelium arises from the LGR5+ SCs and progenitor cells of the first oxyntic gland in the stomach. Quante M. et al. [63] developed a mouse model with permanent overexpression of IL-1β in the esophagus (L2-IL-1β mice). At the age of 12–15 months, these mice showed MUC5AC+ TFF2+ Notch1+ columnar metaplasia of the distal esophagus without goblet cells. After treatment with deoxycholic acid, these mice developed Barrett-like metaplasia, with the expression of мPHK Tff2, Cckbr, Muc5ac, Cdx2, Krt19, Bmp4 and Shh. Inhibition of the Notch signaling pathway led to the acquisition of an intestinal phenotype enriched with goblet cells. These findings provided evidence that the metaplastic epithelium evolves from a cardiac to an intestinal phenotype. Additionally, the cardiac phenotype is due to the LGR5 + SCs and Dclk1 + progenitor cells of the first oxyntic gland, which proliferate and spread to the distal esophagus. The expression of mRNA LGR5 and Dclk1 has also been confirmed in human biopsy specimens through real-time PCR. Further research demonstrated the role of the Notch signaling pathway and its association with NF-κB activation in LGR5+ SCs in the development of metaplasia, dysplasia and EAC [64]. LGR5 expression is elevated in high-grade dysplasia and EAC, thus suggesting that LGR5+ SCs are involved in carcinogenesis [65,66]. The overexpression of LGR5 is associated with poor survival in patients with EAC, independently of their disease stage, age, and treatment with neoadjuvant or adjuvant therapy [66,67].

Lavery D.L. et al. [68] demonstrated that the metaplastic glands of BE are structurally similar to the pyloric glands of the stomach. The proliferative zone in both glands is located at the middle third of the mucosa, and is characterized by the expression of LGR at the bottom and the bidirectional migration of IdU-traced cells. Epithelial cells in the upper compartment express MUC5AC and TFF1, whereas cells in the lower compartment express TFF2 and MUC6. Goblet cells are localized to the upper third of the mucosa, and express MUC2 and TFF3. Jang B.G. et al. [69] suggested that LGR5+ cells with expression of the intestinal SC markers ASCL2, OLFM4 and EPHB2 in areas of intestinal metaplasia in the stomach and the esophagus are the distinct population of SCs that replace pre-existing SCs. 

Residual embryonic SCs [70,71] and bone-marrow-derived multipotent SCs [72,73] have also been suggested as a BE cellular origin, but this possibility is supported by little evidence.

The most established hypothesis of the origin of BE is the transcommitment of multipotent SCs and progenitor cells. However, different progenitor cells might reasonably be involved in the pathogenesis of BE (Figure 3) and might be responsible for all cellular phenotypes of columnar metaplasia, thereby explaining the heterogeneity and the polyclonal and mosaic-like spread of metaplastic glands.

## 4. Repair of Caustic Injury in the Distal Esophagus

Injury to the squamous epithelium of the esophagus by reflux causes the development of erosive esophagitis. Injury repair in the distal esophagus typically proceeds as follows [2]: in the first stage, granulation tissue forms under a cover of fibrinous exudate for protection of the deep tissues, and the granulations are covered by a reparative epithelium without functional properties, such as the secretion of mucus. Next, the defect is covered with lateral growth of adjacent glands/crypts (ulcer-associated mucosal lineage) until the functional epithelium is restored [2,9].

According to this model, the repair of local injury leads to the proliferation of progenitor cells in the squamous epithelium and the first oxyntic gland in the margins of ulceration. Repeated reflux exposure drives the selection of clones that are resistant to these conditions and form a mucus-secreting, cardiac-type epithelium. Subsequently, the columnar epithelium spreads, thereby increasing in the distal esophagus, and undergoes clonal evolution, thereby forming all gland phenotypes of BE. Therefore, similarly to the that in the pyloric gland, the compartmentalization pattern [50,68] may be explained by the typical process of injury healing with the ulcer-associated mucosal lineage, thus resulting in the phenotype of pyloric metaplasia throughout the gastrointestinal tract [8,9,74].

In a surgical rat model of esophagojejunostomy, the migration of SCs from the stomach to the esophagus was disrupted. Two weeks after the operation, ulceration in the distal esophagus was observed near the anastomosis, which was epithelized by the distal margin with immature crypts of the jejunum [7]. The metaplastic crypts show a similar phenotype to that of the epithelium of the small intestine, with the expression of CDX2, villin, CD10 and MUC2, and an absence of the expression of gastric markers (MUC5AC and MUC6), the intestinal marker Das-1 and the squamous marker p63. The re-epithelization of an ulcer causes epithelial-to-mesenchymal transition (positive expression of E-cadherin in epithelial cells of newly developed crypts, and the co-expression of E-cadherin and TWIST in spindle-like cells in stroma at the ulcer margin) and the migration of cells from the jejunum to the distal esophagus. Proliferation of the immature squamous epithelium is observed at the proximal margin of the ulcer.

In cell cultures of BE without dysplasia and dysplastic BE, bile salts also activate epithelial-to-mesenchymal transition (decreased expression of cadherin 1; increased expression of fibronectin 1, vimentin and matrix metalloprotease 2; and increased cell mobility), which is associated with VEGF signaling [75]. Phipps S.M. et al. [76] found that the genes GPS1 and RRM2, which are suppressed by a low pH, cause epithelial-to-mesenchymal transition in BE with high-grade dysplasia and EAC. Therefore, bile salts in the presence of low pH levels induce epithelial-to-mesenchymal transition, and modulate both the reparation of injury at the distal esophagus and invasion in EAC.

## 5. Roles of Cytokine Storm and a Proinflammatory Microenvironment in BE Development

The concept of reflux-induced, cytokine-mediated injury of the mucosa in the distal esophagus was proposed by Souza R.F. et al. [10,11,12,13,77], who found that in rats with esophagoduodenostomy, ulcers appeared several weeks after the operation [77] and thus were not initiated directly by acid and bile reflux via caustic injury. Instead, the authors observed morphological features of reflux esophagitis, with lymphocytic infiltration of the submucosa on the third day after the operation. Subsequently, the lymphocytic infiltration spread into the lamina propria mucosa and into the squamous epithelium. The infiltration of the lamina propria significantly increased at the end of the first week and increased in the epithelium 3 weeks after the operation. The intensity of the infiltration was stable from the third to eighth weeks after the operation. On the third day after the operation, the infiltrate was composed of only CD3+ CD20- T-lymphocytes, but after the seventh day, several neutrophils were also present. Basal cell hyperplasia was observed 1 week after the operation, and papillary hyperplasia developed 2 weeks after the operation and peaked 4 weeks after the operation. In addition, from the fourth week onward, erosions were found in the distal esophagus. The dynamics of morphological changes were associated with the levels of IL-8 in different compartments of the mucosa and submucosa. Several studies have demonstrated the roles of the pro-inflammatory cytokines IL-1β [37,38] and TNFα [78,79] in reflux-induced injury of the esophageal mucosa. These cytokines cause the chemoattraction of immune cells and activation of NF-κB signaling, thus leading to persistent inflammation. Therefore, the pathogenesis of erosive esophagitis is associated with immune-cell-mediated injury caused by the release of proinflammatory cytokines by keratinocytes, and chemoattraction of T-lymphocytes and other immune cells.

This model has been validated in biopsies from patients with severe reflux esophagitis 1–2 weeks after the cessation of proton pump inhibitor therapy. The histological findings include substantial infiltration of the mucosa, predominantly by CD3+ T-lymphocytes; few or absent neutrophils and eosinophils; aggravation of the basal cells and papillary hyperplasia; and increased spongiosis [78]. An immunohistochemical evaluation indicated overexpression of HIF-2α and phosphorylated p65, associated with increased levels of mRNA and the proinflammatory cytokines IL-8, IL-1β, TNF-α, COX-2 and ICAM-1 [79,80].

### NF-κB and IL6/STAT3 Signaling

Bile acid exposure in the presence of acidic pH increases reactive oxygen species in keratinocytes and the metaplastic epithelium in BE [81], thereby leading to HIF-2α stabilization [82], nuclear translocation, binding to HIF-responsive elements, and the triggering of the synthesis and release of proinflammatory cytokines. Therefore, HIF-2α regulates the inflammatory response to reflux injury, in a manner associated with NF-κB signaling via p65 phosphorylation (Figure 4). NF-κB activation in the distal esophagus not only leads to persistent inflammation but also induces the development of intestinal metaplasia via the activation of CDX2, which is crucial for intestinal differentiation. CDX2 contains a binding site for NF-κB and may be a downstream target of NF-κB [12,13,23,33]. Moreover, after exposure to deoxycholic acid, NF-κB directly activates MUC2 expression [83].

The levels of IL-8 and IL-1β rise from erosive esophagitis to BE, and from BE to EAC, and levels of NF-κB simultaneously increase [37], thus leading to epithelial cell proliferation, prevention of apoptosis and promotion of carcinogenesis [36]. Another important cytokine in BE pathogenesis is IL-6. L2-IL-1β/IL-6−/− deficient mice do not develop metaplasia in the distal esophagus [63]. IL-6 is produced in the metaplastic epithelium and causes the activation and translocation of STAT3 to the nucleus and subsequent synthesis of the antiapoptotic proteins Bcl-xL and Mcl-1 [84,85,86]. This signaling pathway favors epithelial cell survival in an environment of aggressive reflux. Moreover, autocrine IL-6 signaling in EAC induces cell proliferation and angiogenesis, which promote cancer progression (Figure 5). The interaction and reciprocal activation of the IL-6/STAT3 and NF-κB pathways lead to persistent inflammation and drive carcinogenesis.

## 6. Genomic Alterations in BE Carcinogenesis

Repeated bile acid exposure in an acidic pH causes oxidative stress in squamous and metaplastic epithelia in BE, thereby leading to DNA damage [24], particularly double-strand breaks [87,88]. Moreover, hydrochloric acid in the lumen of the esophagus reacts with the nitrites in saline, thus causing the release of nitric oxide, which in turn may lead to double-strand breaks in DNA [87,89,90,91]. Double-strand breaks are hazardous, because their repair results in loss of the original DNA sequence, thus contributing to frameshift and truncating mutations, and to the loss of large DNA regions spanning several genes, thereby potentially leading to the loss of heterozygosity.

The mutational load in BE without dysplasia varies widely and has been estimated to be 0.42–1.28 mutations per Mb, or higher [6,92,93,94]. The mutational load increases in the following conditions, from lowest to highest: BE without dysplasia, low-grade dysplasia, high-grade dysplasia, and EAC [95]. In EAC, the mutational load can reach 7.33–9.9 mutations per Mb [94,96,97]. Higher mutational load is associated with a higher risk of neoplastic progression [6,92,98]. However, most studies have examined small series of samples. In a large sample of patients, Eluri S. et al. [93] found no differences in mutational load between progressors and non-progressors.

The most important factors in neoplastic progression in the metaplastic segment are clones with mutations in TP53, which evoke a marked increase in genetic abnormalities. TP53 mutations are observed in 72% of EAC cases [96]. Next-generation sequencing indicated that these mutations may arise years before a histological diagnosis of dysplasia in progressors and are seen in only 2.5–5% of non-progressors [5,6,99].

*TP53* mutations facilitate further genetic mechanisms of carcinogenesis in BE. Loss-of-function mutations in TP53 evoke exponential growth in several mutations, owing to impaired mechanisms of DNA repair and apoptosis. Moreover, the mutated p53 acquires non-canonical functions, such as induction of epithelial-to-mesenchymal transition and activation of NF-κB [100].

Mutations in *TP53* serve as a point from which to choose the direction in the realization of different genetic mechanisms (Figure 6). A classical pathway with a gradual increase in the number of mutations and with consecutive inactivation of suppressor genes *(CDKN2A, SMAD4* and *TP53*) is rarely observed. More often (62.5% of EAC cases), mutations in *TP53* precede rapid whole-genome doubling and chromosome instability with the amplification of oncogenes in cancer cells [5]. A third pathway includes catastrophic genetic events, such as chromothripsis, kataegis and breakage–fusion–bridge cycles, due to impaired DNA reparation [39,97,101].

Chromothripsis—a catastrophic genomic event involving simultaneous large genomic rearrangements, including chromosomal shattering, gains and losses of regions containing several genes—may cause the rapid activation of oncogenes and the inactivation of tumor suppressor genes [102]. Notably, chromothripsis may lead to the amplification of *MYC* and *MDM2* oncogenes [97].

Genetic catastrophes are also associated with *TP53* mutations. Rausch T. et al. [103] showed such an association in children with sonic hedgehog medulloblastoma associated with Li–Fraumeni syndrome. The authors elucidated three mechanisms of chromothripsis associated with *TP53* mutations: (1) critical telomere shortening followed by chromosome end-to-end fusions, (2) premature chromosome compaction due to cell cycle impairment (G2/M transition checkpoint) and (3) impaired DNA repair and apoptosis induction mechanisms. A high frequency of TP53 mutations and telomere shortening may explain the high frequency of chromothripsis in EAC (30–32.5% of cases) [97,101].

Chromothripsis often coexists with kataegis, a pattern of hypermutation with cluster changes in C > T and C > G in TpC dinucleotides, which was first described in breast cancer [104,105]. Kataegis is associated with the activity of apolipoprotein B mRNA-editing enzyme and catalytic polypeptide (APOBEC) protein members. The APOBEC protein family is a group of cytosine deaminases that target nucleic acids and induce C  >  U changes that may lead to mutations [106]. In the cytoplasm, APOBECs prevent the replication of DNA-containing viruses (such as the human immunodeficiency virus) and serve as a component of the innate retroviral defense [107]. APOBECs target single-stranded DNA and produce clusters of strand-coordinated mutations. Kataegis is observed at areas of DNA fragmentation during chromothripsis after telomere crisis [108]. The frequency of kataegis in EAC varies from 31 to 86.4% of cases [97,101].

The breakage–fusion–bridge cycle begins with a loss of telomeres, which is followed by the fusion of chromosome ends or sister chromatid fusion, with the formation of double-minute chromosomes that break during anaphase [109]. This process repeats through several cell cycles, thereby leading to the formation of inverted duplications with the amplification of certain chromosome regions. Malignant transformation is induced when such amplified regions include oncogenes. Breakage–fusion–bridge cycles are observed in 27% of EAC cases, and trigger the amplification of the potent oncogenes *RCF3, MDM2, VEGFA, BCAT1* and *KRAS* [97,101].

These data provide evidence that genomic catastrophe is important in malignant transformation in BE, thus serving as an alternative mechanism of carcinogenesis. The genomic catastrophe frequently observed in high-grade dysplasia and EAC is likely to explain the rapid neoplastic progression in BE [39,94,97,101]. Moreover, catastrophic events are a “point of no return” after which malignant transformation becomes inevitable [110].

## 7. Conclusions

The development of intestinal metaplasia in the distal esophagus is a multistep process that occurs with exposure to bile reflux under an acidic pH, thus triggering the creation of a proinflammatory microenvironment. Exposure to bile acids and cytokine storms causes the reprogramming of SCs and progenitor cells, thereby leading to the development of cardiac metaplasia. Subsequently, via clonal evolution, different cell populations arise along the metaplasia segment. Increased clonal diversity and active mutagenesis with massive genome rearrangements underlie the neoplastic progression to EAC.

## Figures and Tables

**Figure 1 ijms-24-09304-f001:**
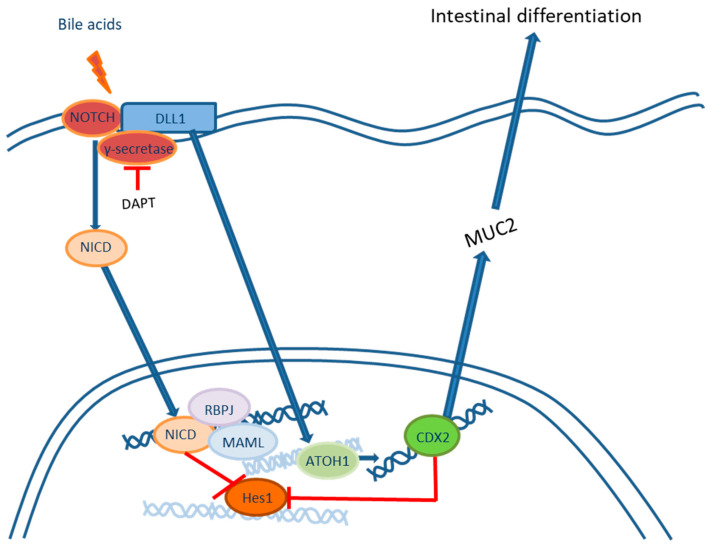
Notch signaling pathway inhibition leads to intestinal differentiation. Blue arrows indicate positive regulation. Red blunt lines indicate negative regulation.

**Figure 2 ijms-24-09304-f002:**
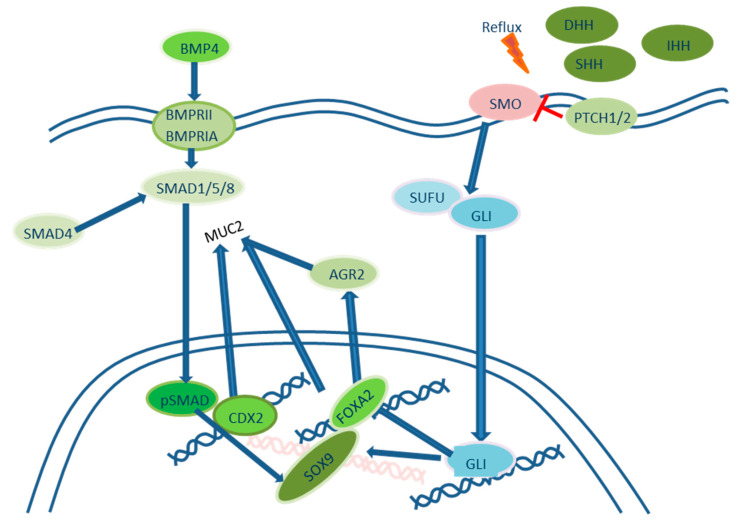
Hedgehog signaling pathway in the development of intestinal metaplasia in the distal esophagus. Blue arrows indicate positive regulation. Red blunt lines indicate negative regulation.

**Figure 3 ijms-24-09304-f003:**
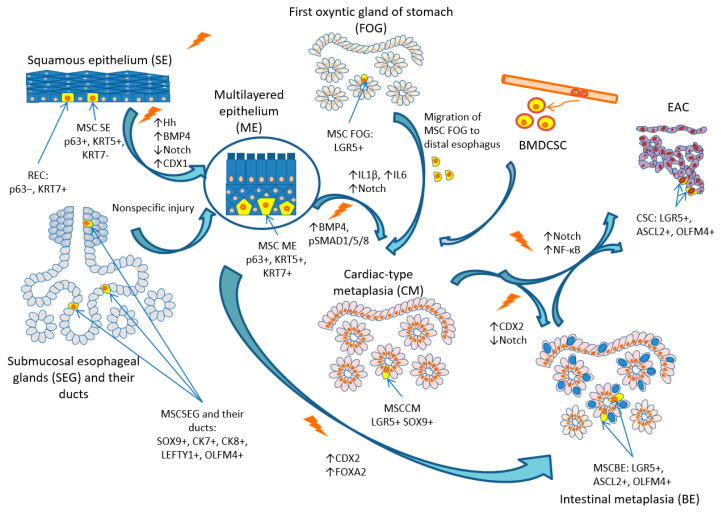
Cellular origins of metaplasia and pathogenesis of BE and EAC. Abbreviations: SE, squamous epithelium; MSCSE, multipotent stem cells of squamous epithelium; REC, residual embryonic cells; ME, multilayered epithelium; MSCME, multipotent stem cells of multilayered epithelium; SEG, submucosal esophageal glands; MSCSEG, multipotent stem cells of submucosal esophageal glands; FOG, first oxyntic gland of the stomach; MSCFOG, multipotent stem cells of the first oxyntic gland of the stomach; BMDCSC, bone-marrow-derived circulating stem cells; CM, cardiac-type metaplasia; BE, Barrett’s esophagus; SCBE, stem cells associated with Barrett’s esophagus; EAC, esophageal adenocarcinoma; CSC, cancer-associated stem cells.

**Figure 4 ijms-24-09304-f004:**
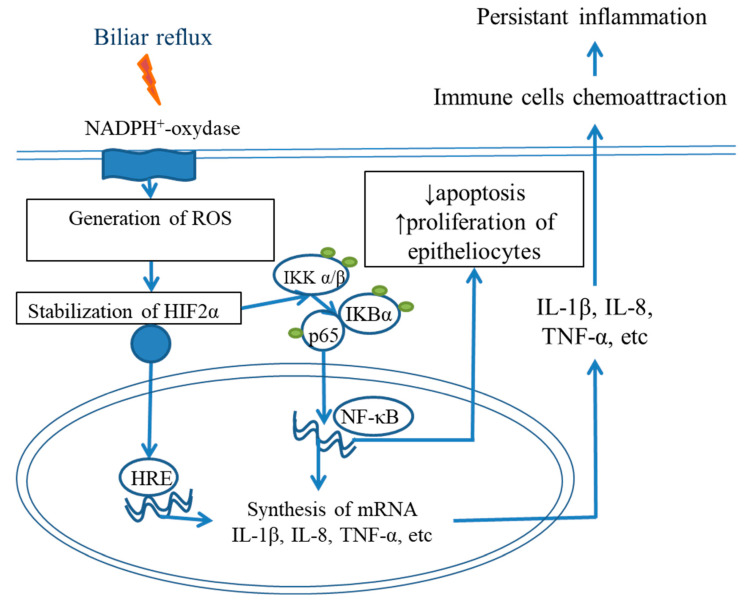
NF-κB signaling pathway in the pathogenesis of reflux esophagitis and BE. Blue arrows indicate positive regulation of a signaling pathway.

**Figure 5 ijms-24-09304-f005:**
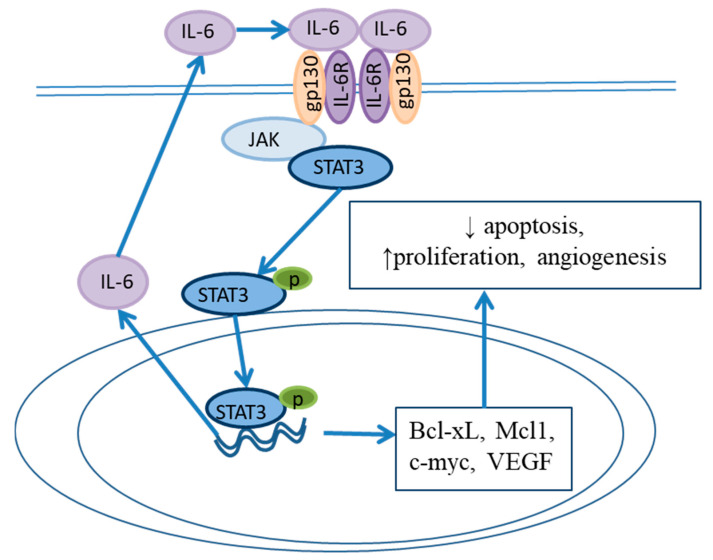
The IL-6/STAT3 signaling pathway in BE. Blue arrows indicate the positive regulation of a signaling pathway.

**Figure 6 ijms-24-09304-f006:**
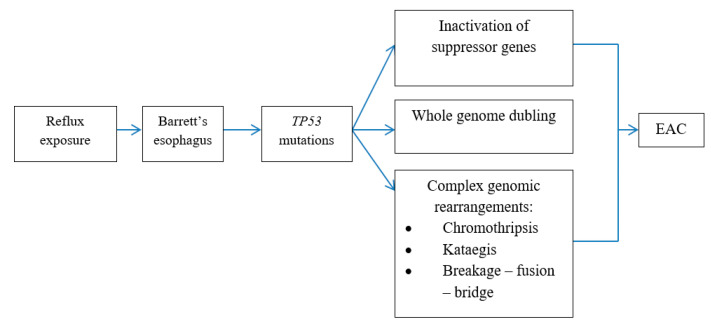
Genetic mechanisms of neoplastic progression in BE.

## Data Availability

Not applicable.

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
