# Peer review of "Signaling Pathways in the Pathogenesis of Barrett’s Esophagus and Esophageal Adenocarcinoma"

_ijms, 2023, doi:10.3390/ijms24119304_

Round 1

Reviewer 1 Report

This review describes the role of bile refluxes and associated signaling pathways, which leads to the development of Barrett’s esophagus (BE) and esophageal adenocarcinoma (EAC). This review will be potentially important for people studying BE and EAC as well as for the general public who wants to know about the effects of acid refluxes on their gastrointestinal health and the occurrence of cancer. However, this review requires better structure and organization to improve its quality before it can be accepted for publication. Here are some major issues, observed in this review.

1. The title and introduction stated that this review “analyzes signaling pathways or molecular pathways in the pathogenesis of BE”. However, the abstract mentioned that “ The review summarizes contemporary concepts of BE and EAC pathogenesis”. The authors need to be consistent with the main idea of this review and they need to organize their text and figures accordingly. All the figures show the signaling pathways involved in BE, but the heading and subheading in the text tell a different story. The authors can mention bile reflux and their effects on BE development in the beginning, then they can put each pathway such as Notch, hedgehog, NF-kB, JAK-STAT etc. as major headings and describe their roles in BE under each section. The roles of all these pathways are mentioned randomly throughout the paper right now, which creates unnecessary confusion for readers. If the same information is presented in a well-organized way, it will help the reader to understand and retain the information more effectively.

2. Similarly, under section 3 “Possible cellular origin of BE”, the authors mention six potential origins of metaplasia in the beginning and then they describe details of each origin in the following paragraphs. In my opinion, it will be better if they can subtitle the six origins and give a description immediately under each subtitle.  

3. The mechanism of BE development should also be divided into two sections, each describing the “repair of caustic injury” and “Cytokine storm” just to be consistent with the information given in the introduction.

4. Figures look fine except that the text inside the boxes is often misaligned. The text needs to be in the middle of the boxes. There are other small mistakes like misaligned “DAPT” in figure 1, FOXA2 is written as “FOX2A” in figure 2.

5. Paragraphs in general are also not organized properly. There are lots of small paragraphs containing 1-3 sentences that can easily be merged into a single paragraph. e.g. paragraph containing lines 151-153 should be part of the above paragraphs, not separate. There are numerous such instances throughout the paper.

6. Sentences are often written in a very confusing manner with highly technical jargon and are often unable to convey the information properly. e.g. sentence written in line 154-156,  “expressional profile of CK7, 8/18, 19 N 20” in line 163, “ mature and immature intestinal phenotypes” in line 31, etc. Figure 6 and line 338 read “Mutation of TP53 induce whole genome doubling”. The whole genome can not be doubled by a mutation. They should replace it with a more appropriate term like “gene duplication” or “genome-wide gene duplication” if they intend to convey that there is gene duplication throughout the genome.

7. Line 253 reads “<<.Cytokine storm>> “. What do the authors mean by special parenthesis around the cytokine storm? Additionally, there is often the use of a letter, which looks like the opposite of the letter “N” in lines 163, 337 etc. Does it mean “and” or something else?

8. Full forms of abbreviation should be mentioned the first time they appear in the text, e.g. LOH appears in line 174 but the full form was written in line 315. 

This review has lots of grammatical errors throughout the paper and needs thorough proofreading. Sentences are written in a very confusing manner. There are many spelling mistakes like hyrdocholic, recidual, heterozygociti, mutataional etc. A sentence can not start with “And” as in lines 184 and 267. The authors should not use abbreviated words like can’t, doesn’t, or didn’t (lines 103, 105, 109, 121, 146, etc). Instead, they should use can not, does not. Commas are often missing e.g. in lines 54, 91,111, 157, 168, 176, 179,181, 192, 221,227, 230,232, 236,244, 246,251, 257, 259, 260,264, 265, 267, 322, 362, 386 etc. There is often too much spacing between words e.g. in lines 19, 46, 53, 104,116, 178,191, 216, 224,257,265,285 etc. 

Author Response

TO REVIEWER 1

The authors thank the referee for a thorough analysis of our manuscript and important suggestions for improving this work.

Comments and Suggestions for Authors

This review describes the role of bile refluxes and associated signaling pathways, which leads to the development of Barrett’s esophagus (BE) and esophageal adenocarcinoma (EAC). This review will be potentially important for people studying BE and EAC as well as for the general public who wants to know about the effects of acid refluxes on their gastrointestinal health and the occurrence of cancer. However, this review requires better structure and organization to improve its quality before it can be accepted for publication. Here are some major issues, observed in this review.

  1. The title and introduction stated that this review “analyzes signaling pathways or molecular pathways in the pathogenesis of BE”. However, the abstract mentioned that “ The review summarizes contemporary concepts of BE and EAC pathogenesis”. The authors need to be consistent with the main idea of this review and they need to organize their text and figures accordingly. All the figures show the signaling pathways involved in BE, but the heading and subheading in the text tell a different story. The authors can mention bile reflux and their effects on BE development in the beginning, then they can put each pathway such as Notch, hedgehog, NF-kB, JAK-STAT etc. as major headings and describe their roles in BE under each section. The roles of all these pathways are mentioned randomly throughout the paper right now, which creates unnecessary confusion for readers. If the same information is presented in a well-organized way, it will help the reader to understand and retain the information more effectively.

Response:

The headline was changed for: “Signaling pathways Notch, hedgehog, NF-kB and IL6/STAT3 in pathogenesis of Barrett’s esophagus and esophageal adenocarcinoma”.  Analyzed pathways were added as subheadings (Lines 71, 96 and  313), because they are part of pathogenesis. Notch and hedgehog signaling was described in section about development of intestinal metaplasia, but NF-kB and IL-6/STAT3 were shown in context of inflammatory microenvironment.  

  1. Similarly, under section 3 “Possible cellular origin of BE”, the authors mention six potential origins of metaplasia in the beginning and then they describe details of each origin in the following paragraphs. In my opinion, it will be better if they can subtitle the six origins and give a description immediately under each subtitle.

Response:

We added subtitles in section 3 “Possible cellular origin of BE” for better structure of the text (lines 151, 170, 178 and 197). Residual embryonic SC and bone marrow-derived multipotent SC were not subtitled because the information about these origins is few and controversial.

  1. The mechanism of BE development should also be divided into two sections, each describing the “repair of caustic injury” and “Cytokine storm” just to be consistent with the information given in the introduction.

Response:

Section 4 “Mechanism of injury repair in distal esophagus: experimental findings” (line 244) was renamed as “Repair a caustic injury in the distal esophagus” as this section is attributed to caustic injury restoration. And section 5 “Cytokine storm provides microenvironment for BE development” (line 281) was renamed as “Cytokine storm and proinflammatory microenvironment in BE development” and is attributed to cytokine storm.

  1. Figures look fine except that the text inside the boxes is often misaligned. The text needs to be in the middle of the boxes. There are other small mistakes like misaligned “DAPT” in figure 1, FOXA2 is written as “FOX2A” in figure 2.

Response:  Figures were revised and edited.

  1. Paragraphs in general are also not organized properly. There are lots of small paragraphs containing 1-3 sentences that can easily be merged into a single paragraph. e.g. paragraph containing lines 151-153 should be part of the above paragraphs, not separate. There are numerous such instances throughout the paper.

Response:  Paragraphs were changes for longer.

  1. Sentences are often written in a very confusing manner with highly technical jargon and are often unable to convey the information properly. e.g. sentence written in line 154-156, “expressional profile of CK7, 8/18, 19 N 20” in line 163, “ mature and immature intestinal phenotypes” in line 31, etc. Figure 6 and line 338 read “Mutation of TP53 induce whole genome doubling”. The whole genome can not be doubled by a mutation. They should replace it with a more appropriate term like “gene duplication” or “genome-wide gene duplication” if they intend to convey that there is gene duplication throughout the genome.

Response:   “Mature and immature intestinal phenotypes” in line 33 were deleted. “Expressional profile of CK7, 8/18, 19 N 20” in line 182 was edited for “expression of the same markers CK7, 8/18, 19 N 20”. “Mutation of TP53 induce whole-genome doubling” in line 374 means that mutation of TP53 is the reason for future genomic alterations that cause whole-genome doubling (term used by Stachler M.D. et al.), not just gene duplication or amplification. The sentence was edited as “More often (62,5% of EAC) mutations of TP53 precede rapid whole genome doubling”.

  1. Line 253 reads “<<.Cytokine storm>> “. What do the authors mean by special parenthesis around the cytokine storm? Additionally, there is often the use of a letter, which looks like the opposite of the letter “N” in lines 163, 337 etc. Does it mean “and” or something else?

Response:  Parenthesis around the cytokine storm and other extra symbols was deleted.

  1. Full forms of abbreviation should be mentioned the first time they appear in the text, e.g. LOH appears in line 174 but the full form was written in line 315.

Response:  Abbreviation LOH was deciphered in the upper line 216.

  1. Comments on the Quality of English Language

This review has lots of grammatical errors throughout the paper and needs thorough proofreading. Sentences are written in a very confusing manner. There are many spelling mistakes like hyrdocholic, recidual, heterozygociti, mutataional etc. A sentence can not start with “And” as in lines 184 and 267. The authors should not use abbreviated words like can’t, doesn’t, or didn’t (lines 103, 105, 109, 121, 146, etc). Instead, they should use can not, does not. Commas are often missing e.g. in lines 54, 91,111, 157, 168, 176, 179,181, 192, 221,227, 230,232, 236,244, 246,251, 257, 259, 260,264, 265, 267, 322, 362, 386 etc. There is often too much spacing between words e.g. in lines 19, 46, 53, 104,116, 178,191, 216, 224,257,265,285 etc.

Response:  Spelling mistakes and extra spacing were edited.

Reviewer 2 Report

Maslenkina et al wrote a review about the pathogenesis of esophageal adenocarcinoma (EAC) development from Barrett esophagus (BE). Main comments:

1) The main problem is that this paper is very hard to read. Several sentences are badly written and difficult to understand.

2) In figures, it is unclear whether arrows represent inhibitory or stimulatory signalling.

3) The review does not take into account the possible role of obesity, which is frequently associated with BE, as a risk factor, considering all the cytokines produced by adipose tissue.

4) Another risk factor that was not analyzed in BE is smoking.

Very bad

Author Response

TO REVIEWER 2

The authors thank the referee for a thorough analysis of our manuscript and important suggestions for improving this work.

Comments and Suggestions for Authors

Maslenkina et al wrote a review about the pathogenesis of esophageal adenocarcinoma (EAC) development from Barrett esophagus (BE). Main comments:

1) The main problem is that this paper is very hard to read. Several sentences are badly written and difficult to understand.

Response:  Long sentences were broken into smaller. Mistakes in English language were edited.

2) In figures, it is unclear whether arrows represent inhibitory or stimulatory signalling.

Response:   Arrows show positive regulation in signaling pathway. Lines with a blunt end indicate inhibition.

3) The review does not take into account the possible role of obesity, which is frequently associated with BE, as a risk factor, considering all the cytokines produced by adipose tissue.

Response:  We analyzed local proinflammatory microenvironment in esophageal mucosa and submucosa, that is caused by biliary reflux, not general proinflammatory state as in obesity. Adipokines and particularly leptin was not the focus of our review.

4) Another risk factor that was not analyzed in BE is smoking.

Response:  The aim of our review was not to cover all the risk factors for development of BE, but to focus on several pathways in pathogenesis of BE and EAC. 

5) Comments on the Quality of English Language

Very bad

Response:  English language was revised.

Round 2

Reviewer 1 Report

The authors have made all the changes I requested. I feel that the structure of the review has improved a lot, hence I recommend this review to be accepted. 

The authors have made necessary changes in the language. 

Author Response

TThe authors thank the reviewer for carefully reading our manuscript and for valuable comments.

Reviewer 2 Report

I still think that the overall quality is not enough 

see abobe

Author Response

I thank the Reviewer for his efforts.